# Isoniazid resistance profile and associated levofloxacin and pyrazinamide resistance in rifampicin resistant and sensitive isolates/ from pulmonary and extrapulmonary tuberculosis patients in Pakistan: A laboratory based surveillance study 2015-19

**Sabira Tahseen**[1,2]*, **Faisal Masood Khanzada**[1], **Alamdar Hussain Rizvi**[1], **Mahmood Qadir**[1], **Aisha Ghazal**[1], **Aurangzaib Quadir Baloch**[3], **Tehmina Mustafa**[2,4]

**1** National TB Reference Laboratory, National TB Control Program, Islamabad, Pakistan, **2** Centre for International Health, Department of Global Public Health and Primary Care, University of Bergen, Bergen, Norway, **3** National TB Control Program, Ministry of National Health Services Regulation and Coordination, Islamabad, Pakistan, **4** Department of Thoracic Medicine, Haukeland University Hospital, Bergen, Norway

* sabira.tahseen@gmail.com

## Abstract

### Background

Pakistan is among top five high burden countries for tuberculosis and drug resistant TB. Among rifampicin sensitive new pulmonary TB (PTB), prevalence of isoniazid resistance is 8.3% (95%CI: 7.0–10.7) and resistance to fluoroquinolone is higher (11·1%, 95%CI: 7·8–14·3) than isoniazid resistance.

### Method

Five year retrospective data (2015–2019) of drug susceptibility testing (DST) for *Mycobacterium tuberculosis* isolates, performed using recommended phenotypic (pDST) and/or genotypic (gDST) methods was analyzed stratified by rifampicin results for isoniazid resistance profiles and associated levofloxacin and pyrazinamide resistance.

### Findings

DST data was analyzed from 11045 TB patients. Isolates were tested using pDST (87%), gDST (92%) and both methods (79.5%). For both rifampicin and isoniazid, a significant difference (*P* < .001) was noted between resistance detected by pDST and gDST. Among isolates, tested by both methods (8787), 49% were resistant to rifampicin and 51.7% to isoniazid with discordance in resistant results of 15.8% for each, with 13.2% (570) of rifampicin resistance reported sensitive by pDST and 14.2% (660) of isoniazid resistance missed by gDST. Estimated isoniazid resistance among rifampicin sensitive new PTB, extrapulmonary TB and previously treated PTB was 9.8% (95%CI: 8.7–11.1), 6.8% (95%CI: 5.4–

**Data Availability Statement:** All relevant data are within the manuscript and its Supporting Information files.

**Funding:** The author(s) received no specific funding for this work.

**Competing interests:** The authors have declared that no competing interests exist.

8.5) and 14.6% (95%CI: 11.8–17.9) respectively. Significant differences were reported between the genotypic profile of isoniazid resistance associated with rifampicin-resistant and sensitive isolates including detectable mutations (87% vs 71.6%), frequency of *inhA* (7.6% and 30.2%) *and katG* mutations (76.1% vs 41.2%) respectively. Among rifampicin resistant and sensitive isolates, a significantly higher level of resistance to levofloxacin and pyrazinamide was seen associated with isoniazid resistance.

## Conclusion

There are risks and many challenges in implementing WHO recommended treatment for isoniazid resistant tuberculosis. The laboratory based surveillance can complement random surveys in country specific planning for TB diagnostics and appropriate treatment regimens.

## Introduction

Isoniazid (INH) is one of the most important first-line medicines for the treatment of active tuberculosis (TB) with high bactericidal activity and a good safety profile [1–3]. Together with rifampicin (RMP), the two drugs represent the cornerstone of World health organization (WHO) recommended first-line treatment regimen used worldwide [4]. INH is also used in high dose in short course second line treatment regimens for drug resistant TB (DRTB) [5]. INH is critical not only for the treatment of active TB, but it is also highly effective in preventing disease and is the most commonly used medicine for latent TB infection [6]. INH resistance can thus undermine the effectiveness of treatment for both TB disease and infection.

Culture-based phenotypic testing is the current reference method for testing anti-TB medicines. It relies on testing at critical concentrations of drugs, that is, the lowest concentration of an anti-TB medicine that inhibits the in vitro growth of 99% of phenotypically wild-type strains of *Mycobacterium tuberculosis* (Mtb) [7]. In 2008, WHO endorsed the first genotypic drug susceptibility testing (DST) method for the rapid detection of multidrug-resistant (MDR) TB. INH resistance has been associated with multiple genes, most frequently *katG* and *inhA* [8–11]. The reported frequency of mutations varies in different geographical regions [11]. A number of studies have reported that the mutation at codon 315 in *katG* gene is often associated with a high level of INH resistance whereas mutations in *inhA* gene are associated with low-level resistance [8, 9].

Globally there are an estimated 10 million incident TB cases and among these, 7.0 million were notified in 2018. A high treatment success rates of at least 85% for new TB cases is regularly reported by all countries [7]. The estimated global prevalence of INH resistance among RMP sensitive new and previously treated TB (RsHr-TB) is 7.4% (95%CI: 6.5%–8.4%) and 11.4% (95%CI:9.4%–13.4%) respectively [12]. People infected with a TB strain that is resistant to INH, are reported to have a higher rate of unfavorable treatment outcomes with standard first line treatment [13].

In 2019, WHO issued guidelines recommending a modified 6-month treatment regimen containing RMP, ethambutol (EMB), pyrazinamide (PZA) and levofloxacin (LFX) for people with RsHr-TB. Exclusion of resistance to RMP is strongly recommended and empirical treatment is not advised. Fluoroquinolone (FQ) and PZA testing prior to treatment, is also advised where possible in order to prevent the acquisition of additional drug resistance. In addition, information on the specific INH mutations (*katG or inhA*) and overall host acetylator status at country or regional level are considered useful for regimen design [5].

Pakistan is country in South Asia with 212 M population and is among top five high burden countries (HBC) for TB and drug resistant TB (DRTB) [7]. TB disease burden is estimated at 562K incident TB cases at 265 /100K population. DOTs is the official strategy for TB control in the country and six month treatment regimen containing RMP throughout was adopted starting from 2012. Treatment success rate is maintained above 90% for new TB patients. In 2018, 64% of the estimated TB cases were notified including 80% pulmonary TB (PTB) [7]. First population based drug resistance survey (DRS) for smear positive PTB patients was conducted in 2012–13 and prevalence of RMP resistance was estimated at 4.2% and MDR at 3.7% in new cases. Any resistance to INH not associated with RMP resistance (RsHr-TB) was reported in 8.3% (95%CI: 7.0–10.7) of new and 7.1% (95%CI: 4.0–11.4) in previously treated PTB patients [14]. Subsequently, DRS isolates were also tested for FQ and PZA resistance among RMP resistant and sensitive population as part of a multicounty surveillance project and FQ resistance was reported respectively in 21·8% (95%CI;13·1–30·5)and 11·1%(95%CI:7·8–14·3) and PZA resistance in 39.5% (30.1–48.9) and 0.5%(0.1–0.8) [14, 15]. However there is limited published data on molecular markers for INH resistance [12]. We analyzed routine laboratory data to study diagnostic and clinical implications of the prevalent phenotypic and genotypic profile of INH resistance and LFX and PZA resistance associated with INH resistance in RMP resistant (RrHr) and sensitive (RsHr) isolates from pulmonary and extrapulmonary TB (EPTB) patients.

## Study setting, design and methodology

### Study setting

In Pakistan, Xpert MTB/RIF (Cepheid, Sunnyvale, CA, USA) testing services are decentralized, patient reported to have RMP resistance are referred to specialized DRTB treatment sites and specimens are than referred for DST. Specimen transport is well established between DRTB treatment sites and culture and DST laboratories. National TB Reference laboratory (NTRL) is located in Islamabad, the federal capital of the country and receive clinical specimens or culture isolates, referred routinely from treatment sites across Punjab province and three territories including Islamabad, Azad Jammu Kashmir and Gilgit Baltistan together covering more than 50% of the country population. NTRL offer diagnostic and DST services for patients already diagnosed as RMP resistant by Xpert MTB/RIF or who are at risk of drug resistance or presumed to have TB specially PTB in children and EPTB.

### Study design

This is a retrospective five-year (January 2015-December 2019) laboratory based surveillance study. All confirmed Mtb isolates from patients having PTB or EPTB, tested either using phenotypic DST (pDST) and/or genotypic method (gDST) with DST results reported for both RMP and INH were included.

### Laboratory methods

Throughout the study period, gDST for RMP, INH and FQ was performed by Line Probe assay(LPA), using GenoType MTBDRplus and MTBDRsl version 2.0 (Hain Lifescience, Nehren, Germany). MGIT 960 automated system (BD, Sparks, MD, USA) was used to perform pDST at recommended critical concentrations for RMP (1.0ug/ml), INH (0.1ug/ml), PZA (100ug/ml) and ofloxacin (2ug/ml) during 2015–17 and LFX (1.0ug/ml) during 2018–19 [16].

All clinical samples were processed for culture and pDST was performed for all confirmed Mtb isolates. LPA was introduced in late 2015 and gDST was increasingly performed directly

on clinical samples from known RMP resistant patients. Culture isolates were used for gDST for patients having RMP resistance with invalid results on direct testing, known RMP sensitive or with unknown RMP status. Sequencing was not performed as facilities were not established.

All recommended quality control measures for DST were followed during study period including, testing of known sensitive and resistant control strains with each batch of DST [17] and regular participation in annual external quality assessments conducted by WHO collaborating center (Supra National TB reference laboratory, ITM, Antwerp, Belgium) with successful certification for first and second line DST.

## Data management

Case based data was extracted from computerized laboratory information system of NTRL in CSV format. Data was then checked for duplications, for each patient only first pDST and /or gDST results reported either from same or paired samples were included. Duplicate or sequential DST results from same patient were excluded. After cleaning, all personal identifiers were removed before analysis.

## Data analysis

Data were analyzed using STATA1 v13.1 (StataCorp, 4905 Lakeway Drive, College Station, Texas 77845, USA). Mean, median and quartiles were analyzed for quantitative variables and 95% confidence intervals for comparisons between groups. Two sample proportion test was used to analyze differences in proportions between groups. A p-value of <0.05 was considered statistically significant.

Study population was analyzed for demographic characteristics of patients, previous history of TB treatment, province of residence, referring health facilities and AFB smear results. All phenotypic and genotypic results were first analyzed independently for proportion of RMP and INH resistance. Further analysis was done on large subset of Mtb isolates tested by both DST methods. LPA results were interpreted based on recommended guidelines [18]. Results were compared for agreement and discordance between two DST methods for RMP and INH. For final interpretation, resistance conferring mutations to RMP and INH were considered to be true resistance, even if phenotypic testing showed susceptibility. Based on the final interpretations, prevalence of INH and RMP resistance, genetic profile of INH resistance and associated resistance to LFX and PZA was analyzed stratified by RMP result, disease site and previous history of TB treatment. Genetic profiles of INH resistant isolates was studied for the frequency of mutations known to confer resistance among samples displaying either phenotypic and/or genotypic resistance to INH. The INH resistant isolates were sorted into four groups: Group1: phenotypic resistant but genotypic wild type (WT), Group 2: isolates with *inhA* mutation/s only, Group 3: isolates with *katG* mutation only and Group 4: isolates with combined *katG* and *inhA* mutations. Annual trends were analyzed for RsHr-TB, genetic profile of INH resistance and associated FQ and PZA resistance.

Patient having no or unknown previous history of ATT were categorized as new and those with history of previous treatment or on treatment for more than a month as previously treated. For estimation of INH resistance among previously treated, data was analyzed only from those patient with history of WHO recommended TB treatment for new TB patients. Annual trend for INH resistance was analyzed for years with DST results available by both methods for more than 100 patients. For FQ resistance, all strains reported resistant to OFX (2015–17), LFX (2018–19) or showing FQ conferring mutations on LPA were considered LFX resistant. Results were compared with the prevalence estimates of resistance reported in DRS [14] and primary drug resistance in EPTB [19].

### Ethics statement

The study protocol was approved by IRB of Common unit for HIV/AIDS, TB and Malaria program, Islamabad, Pakistan. The antimicrobial resistance was analyzed in Mtb strains isolated routinely in the laboratory for diagnostic purposes. To maintain confidentiality of the patients, de-identified data was used for analysis.

## Results

### Study population

During the study period (January 2015 to December 2019), altogether 11,680 DSTs were reported, 635 were duplicate and/or sequential DST, which were excluded, and final analysis was performed on 11,045 DST results from same number of patients (Fig 1). Patients included were referred from 86 health facilities in 29 districts. Altogether 65.8% patients were referred by tertiary care, 30.2% by secondary and 3.9% by primary health care facilities and 83.8% of patient referral was from health facilities managing DRTB (S1 Table).

Median age of TB patients was 30 years (32 years for PTB and 23 years for EPTB), 7.5% were children (<15yrs), 48.8% females and 86% were resident of Punjab province. Of all patients, 87.3% presented with PTB, 56% of PTB and 8.3% of EPTB patients had a history of previous TB treatment. Among previously treated PTB, 44.8% had history of TB treatment regimen for new patients, 11.8% for retreatment and 22.4% for DRTB (S1 Table). Details of pulmonary and extrapulmonary specimens processed for testing is given in S2 Table.

### Rifampicin and isoniazid resistance

**Rifampicin and isoniazid resistance by DST methods.** Of all 11045 TB patients, pDST results were available for 9620 including 14.1% (1352) EPTB, gDST for 10212 including 12.6% (1282) EPTB and both pDST and gDST results for 8787 patients including 14.1% (1236) EPTB. RMP and INH resistance detected by pDST and gDST is shown in Table 1 and annual trend in S3 Table. Among PTB isolates, a significant difference ($P < .001$) was seen between resistance detected by pDST and gDST for RMP (49.8 vs 56.4%) as well as INH (58.4 vs 49.2%) respectively but difference was not significant among EPTB isolates (Table 1).

**Correlation between genotypic and phenotypic DST results.** Of all isolates tested by both DST methods (n = 8787), an agreement in results was reported for both RMP and INH in 81% (7117), RMP in 92.3% (8109) and INH in 91.9% (8071) of isolates (Table 2; S3 and S4 Tables). A significant difference was seen between RMP resistance ($P < .001$) detected by pDST (42.5%, 95%CI; 41.5–43.5) and gDST (47.7%, 95%CI; 46.7–48.8) and INH resistance ($P < .001$) detected by pDST (51.1%, 95%CI; 50.0–52.1) and gDST (44.2%:95%CI 43.1–45.2). Compared to pDST proportion of INH resistance by gDST was significantly lower among RMP resistant (97.3 vs 83.1%) and RMP sensitive (16.9 vs 8.7%) isolates. Taking both DST results into account, altogether RMP resistance was detected in 4303 (49%, 95%CI; 47.9–50.0) isolates with 678 (15.8%) discordant results including 570(13.2%) reported sensitive by pDST and 108(2.5%) by gDST. INH resistance was detected in 4542(51.7%, 95%CI; 50.6–52.7) isolates with 716(15.8%) discordant result including 660(14.2%) reported sensitive by gDST and 56 (1.2%) by pDST. (Table 2; S4 Table).

**Isoniazid resistance in rifampicin sensitive TB.** Among RMP sensitive new PTB (n = 2489) and EPTB (n = 1058) patients, INH resistance (RsHr-TB) was reported in 9.8% (95%CI 8.7–11.1) and 6.8% (95%CI 5.4–8.5) respectively and 14.6% (95%11.8–17.9) among patients treated previously for new TB (n = 547) (Table 3). A stable annual trend of RsHr-TB was seen in new PTB and EPTB patients (Fig 2). A significant difference was reported in

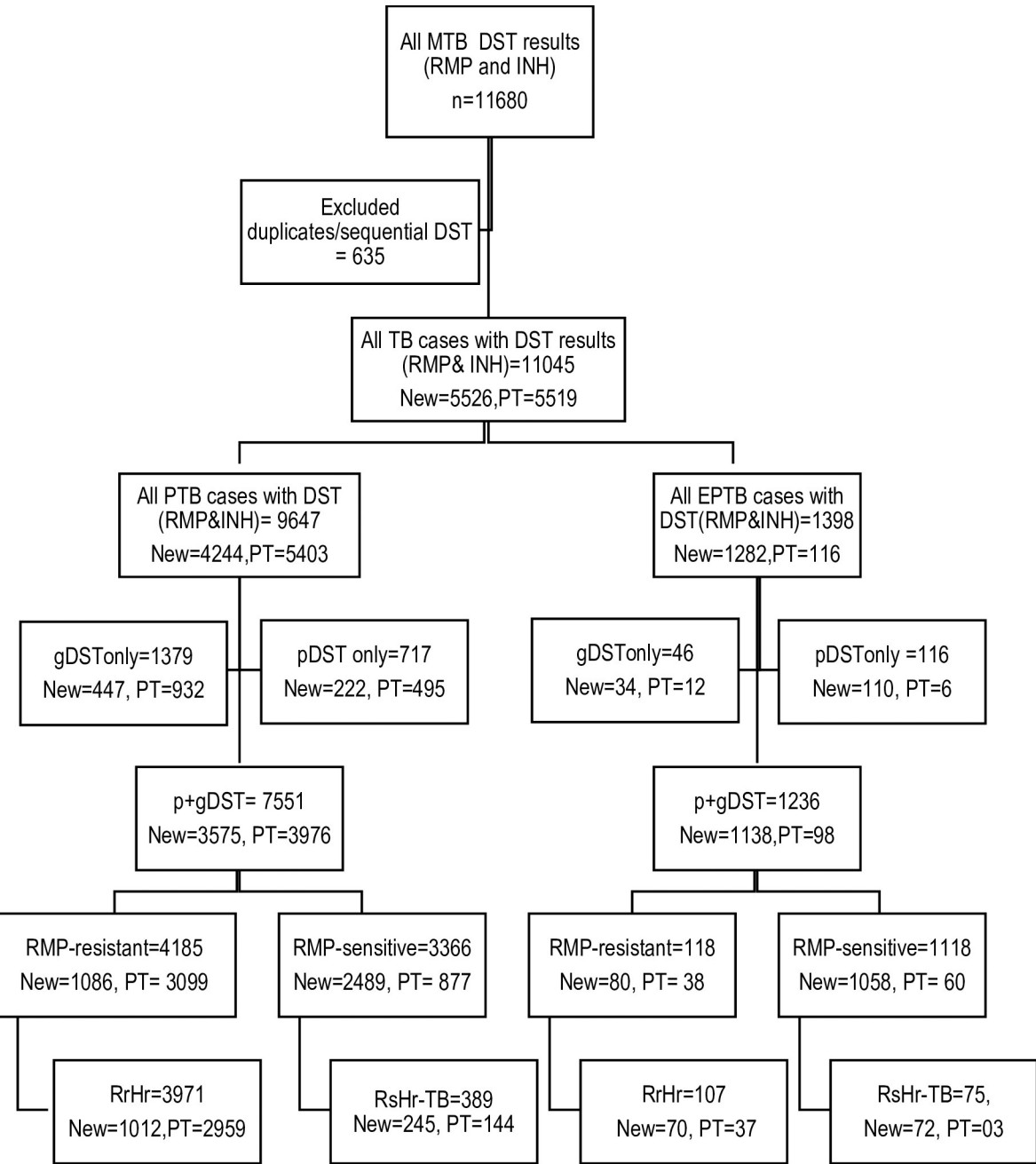

**Fig 1. Flow diagram showing drug susceptibility testing of *Mycobacterium tuberculosis* isolates by disease site, history of TB treatment and DST method, National TB reference laboratory, Pakistan 2015–2019.** RMP-rifampicin, INH-isoniazid, R-resistant, S-sensitive, pDST-phenotypic drug susceptibility testing; gDST-Genotypic drug susceptibility testing, Rr-rifampicin resistant, Rs-rifampicin sensitive, Hr-isoniazid resistant, Hs-Isoniazid sensitive.

proportion of RsHr-TB between new PTB and EPTB ($P$ = 0.004) and new and previously treated PTB patients $(P<0.01)$. Among new PTB, significant differences were not seen between children and adults, male and females and resident of Punjab and other regions (Table 3).

**Table 1. Isoniazid and rifampicin resistance detected by phenotypic and genotypic DST methods in Mtb isolates from pulmonary and extrapulmonary TB patients, National TB reference laboratory, Pakistan 2015–19.**

| Isolates tested | All TB (n = 11045) | | | Pulmonary TB (n = 9647) | | | Extrapulmonary TB (n = 1398) | | |
|---|---|---|---|---|---|---|---|---|---|
| | pDST | gDST | p+gDST | pDST | gDST | p+gDST | pDST | gDST | p+gDST |
| | 9620 | 10212 | 8787 | 8268 | 8930 | 7551 | 1352 | 1282 | 1236 |
| **Any rifampicin resistance** | | | | | | | | | |
| RMP resistant-n | 4225 | 5146 | 4303 | 4117 | 5027 | 4185 | 108 | 119 | 118 |
| RMP-resistant-% | 43.9% | 50.4% | 49.0% | 49.8% | 56.3% | 55.4% | 8.0% | 9.3% | 9.5% |
| (95%CI) | (42.9–44.9) | (49.4–51.4) | (47.9–50.0) | (48.7–50.9) | (55.2–57.3) | (54.3–56.5) | (6.6–9.6) | (7.7–11.0) | (8.0–11.3) |
| **Any isoniazid resistance** | | | | | | | | | |
| INH resistant-n | 5025 | 4557 | 4542 | 4828 | 4393 | 4360 | 197 | 164 | 182 |
| INH resistant-% | 52.2% | 44.6% | 51.7% | 58.4% | 49.2% | 57.7% | 14.6% | 12.8% | 14.7% |
| (95%CI) | 51.2–53.2 | 43.7–45.6 | 50.6–52.7 | 57.3–59.5 | 48.2–50.2 | 56.6–58.9 | 12.7–16.6 | 11.0–14.7 | 12.8–16.8 |
| **Isoniazid resistance associated with rifampicin resistance (MDR)** | | | | | | | | | |
| RrHr/Rr-TB-n | 4117/4225 | 4112/5146 | 4078/4303 | 4010/4117 | 4008/5027 | 3971/4185 | 107/108 | 104/119 | 107/118 |
| RrHr/Rr-TB % | 97.4% | 79.9% | 94.8% | 97.4% | 79.7% | 94.9% | 99.1% | 87.4% | 90.7% |
| (95%CI) | (96.9–97.9) | (78.8–81.0) | (94.1–95.4) | (96.0–97.9) | (78.6–80.8) | (94.2–95.5) | (94.9–100) | (80.0–92.8) | (83.9–95.2) |
| **Isoniazid resistance associated with rifampicin susceptible (RsHr)** | | | | | | | | | |
| RsHr/Rs-TB-n | 908/5395 | 445/5066 | 464/4484 | 818/4151 | 385/3903 | 389/3366 | 90/1244 | 60/1193 | 75/1118 |
| RrHr/Rs-TB-% | 16.8% | 8.8% | 10.3% | 19.7% | 9.9% | 11.6% | 7.2% | 5.2% | 6.7% |
| (95%CI) | (15.8–17.9) | (8.0–9.6) | (9.5–11.3) | (18.5–20.9) | (8.9–10.8) | (10.5–12.7) | (5.9–8.8) | (1.7–6.8) | (5.4–7.2) |

N = number isolates tested, n = number resistant, pDST = Phenotypic drug susceptibility testing, gDST = Genotypic drug susceptibility testing

## Molecular characterization of isoniazid resistant isolates

Among all RrHr-TB isolates, INH conferring mutations were detected in 87.1% (95%CI: 86.0–88.1) compared to 71.6% (95%CI: 67.2-75-6) in RsHr-TB (*P* < .001). A significant difference (P < .001) was also seen in frequency distribution for both *katG* (76.1% vs 41.2%) and *inhA* (7.6% and 30,2%) mutations between RrHr-TB and RsHr-TB respectively (Table 4).

**Table 2. Correlation between phenotypic and genotypic drug susceptibility testing results for rifampicin and isoniazid in Mtb isolates from all type TB patients, National TB reference Laboratory Pakistan 2015–19.**

| | | All | Rifampicin phenotypic(MGIT960) and genotypic(LPA) DST results | | | | | | | |
|---|---|---|---|---|---|---|---|---|---|---|
| | | | pRgR | pRgS | pSgR | pSgS | pRgNA | pSgNA | pNAgR | pNAgS |
| Isoniazid phenotypic(MGIT960)and genotypic (LPA) DST results | All | 11045 | 3625 | 108 | 570 | 4484 | 492 | 341 | 951 | 474 |
| | pRgR | 3826 | 3097 | 66 | 352 | 311 | | | | |
| | pRgS | 660 | 431 | 38 | 59 | 132 | | | | |
| | pSgR | 56 | 9 | | 26 | 21 | | | | |
| | pSgS | 4245 | 88 | 4 | 133 | 4020 | | | | |
| | pRgNA | 539 | | | | | 485 | 54 | | |
| | pSgNA | 294 | | | | | 7 | 287 | | |
| | pNAgR | 675 | | | | | | | 628 | 47 |
| | pNAgS | 750 | | | | | | | 323 | 427 |
| Isoniazid Resistant | RrHr% | | 97.6% | 96.3% | 76.7% | | 98.6% | | 66.0% | |
| | RsHr% | | | | | 10.3% | | 15.8% | | 9.9% |
| | 95%CI | | 97.0–98.0 | 90.8–99.0 | 73.0–80.0 | 9.5–11.3 | 97.1–99.4 | 12.1–20.2 | 62.9–69.0 | 7.4–13.0 |

p-phenotypic, g-genotypic, DST-drug susceptibility testing, S-sensitive, R-Resistant, NA-not available.

**Table 3. Isoniazid resistance among rifampicin sensitive new and previously treated pulmonary and extrapulmonary TB patients, national TB reference laboratory, Pakistan 2015–2019.**

| | Pulmonary TB | | | Extrapulmonary TB | | |
|---|---|---|---|---|---|---|
| | **New** | **PT*** | **p-value** | **New** | **PT*** | **p-value** |
| **All Patients** | 245/2489;9.8% | 80/547;14.6% | <0.01 | 72/1058;6.8% | 3/60;5.0% | 0.588 |
| | (8.7–11.1) | (11.8–17.9) | | (5.4–8.5) | (1.0–13.9) | |
| **Gender** | | | | | | |
| Male | 119/1263;9.4% | 45/290;15.5% | 0.002 | 28/533;5.3% | 1/31;3.2% | 0.608 |
| | (7.9–11.2) | (11.5–20.2) | | (3.5–7.5) | (0.08–16.7) | |
| Female | 126/1226;10.3% | 35/257;13.6% | 0.122 | 44/525;8.4% | 2/29;6.9% | 0.776 |
| | (8.6–12.1) | (9.7–18.4) | | (6.2–11.1) | (0.8–22.8) | |
| **Age Group** | | | | | | |
| Children(<15yrs) | 36/302;11.9% | 7/31;22.6% | 0.09 | 11/184;6.0% | 0/6;0.0% | 0.536 |
| | (8.5–16.1) | (10.0–41.1) | | (3.0–10.4) | (0.0–45.9) | |
| Adult | 204/2147;9.5% | 72/508;14.2% | 0.002 | 61/869;7.0% | 3/54;5.6% | 0.694 |
| | (8.3–10.8) | (11.3–17.5) | | (5.4–8.9) | (1.1–15.4) | |
| Age NA | 5/40;12.5% | 1/8;12.5% | | 0/5;0.0% | 0/0 | |
| | (4.2–26.8) | (0.03–52.7) | | 0.00% | 0.00% | |
| **Place of Residence** | | | | | | |
| Punjab province | 211/2041;10.3% | 64/446;14.3% | 0.015 | 61/871;7.0% | 3/48;6.3% | 0.853 |
| | (9.1–11.7) | (11.2–18.0) | | (5.4–8.9) | (1.3–17.2) | |
| Outside Punjab | 34/448;7.6% | 16/101;15.8% | 0.01 | 11/187;5.9% | 0/12;0.0% | 0.387 |
| | (5.3–10.4) | (9.3–24.4) | | (3.0–10.3) | | |
| **AFB smear Results** | | | | | | |
| Positive | 95/903;10.5% | 65/385;16.9% | 0.001 | 8/128;6.3% | 0/17;0.0% | 0.287 |
| | (8.7–12.7) | (13.3–21.0) | | (2.7–11.9) | (0.0–19.5) | |
| Negative | 41/516;7.9% | 7/86;8.1% | 0.949 | 19/303;6.3% | 2/34;5.9% | 0.927 |
| | (5.9–10.6) | (3.3–16.1) | | (3.8–9.6) | (0.7–19.7) | |
| Not available | 109/1070;10.2% | 8/76;10.5% | 0.934 | 45/627;7.2% | 1/9;11.1% | |
| | (8.4–12.2) | (4.7–19.7) | | (5.3–9.5) | (0.2–48.2) | |

Number shown are number isoniazid resistant/ number rifampicin susceptible isolates tested, % isoniazid resistant (95%CI)

*PT; previously treated with treatment regimen recommended for new TB patients

**Rifampicin resistant TB.** Among 3971 RrHr-PTB isolates, 13.1% (519) were wild type (WT) on LPA. Genetic mutations associated with INH resistance were detected in *inhA* in 7.6% and *katG* in 76.1%, and combined *katG* and *inhA* mutations in 3.1%. Among 107 RrHr-EPTB isolates, 8.4% (9) were genotypic WT, mutations were detected in *inhA* in 10.3%, *katG* in 79.4%, and double mutation in 2 isolates (1.9%) only. A significant difference was not seen between PTB and EPTB (Table 4), new and previously treated (S5 Table) and annual trends (Fig 3; S6 Table).

Among all MDR isolates (PTB and EPTB), the most frequent mutation were reported in codon S315T1 (97.2%, 3028/3112) in *katG* gene and *C-15T* in *InhA* promoter region (92.3%, 287/311) (Table 4).

**Rifampicin sensitive TB.** Among 389 RsHr-PTB isolates, 29.6% (115) were genotypic WT, INH conferring mutations were detected in *inhA in* 29.6%, *katG* gene in 40.6% and double mutation in only one isolate. Among 75 RsHr-EPTB isolates, 22.7% (17) were WT on LPA, INH conferring mutations were detected in *inhA* in 33.3%, *katG* in 44%, with no double mutation. Significant differences were not noted between isolates from PTB and EPTB (Table 4),

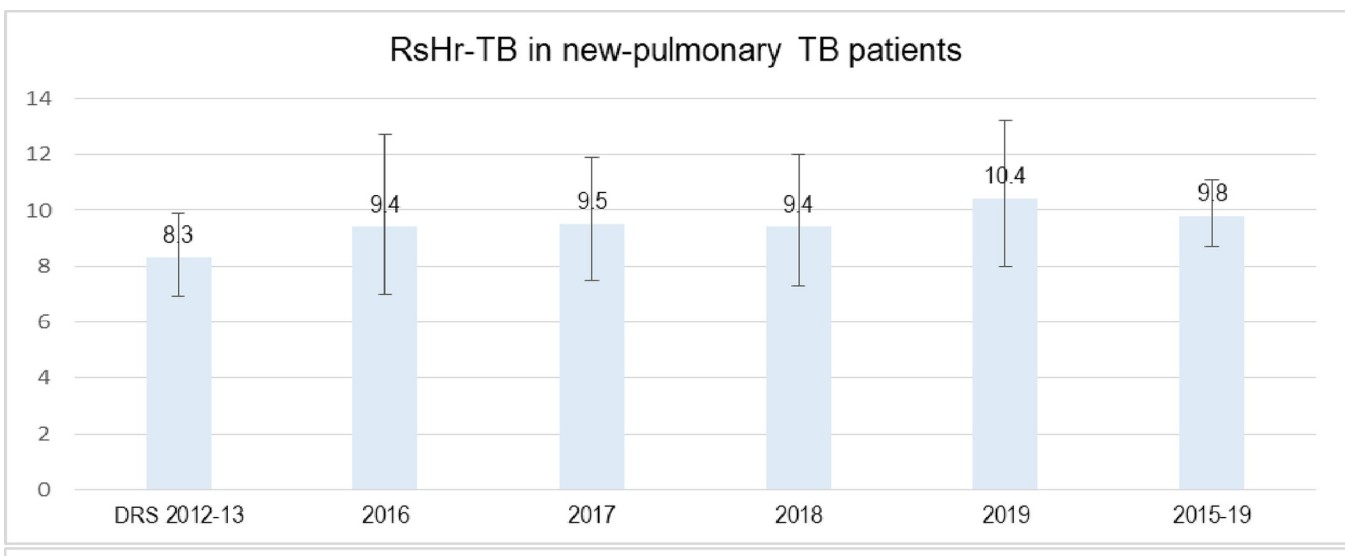

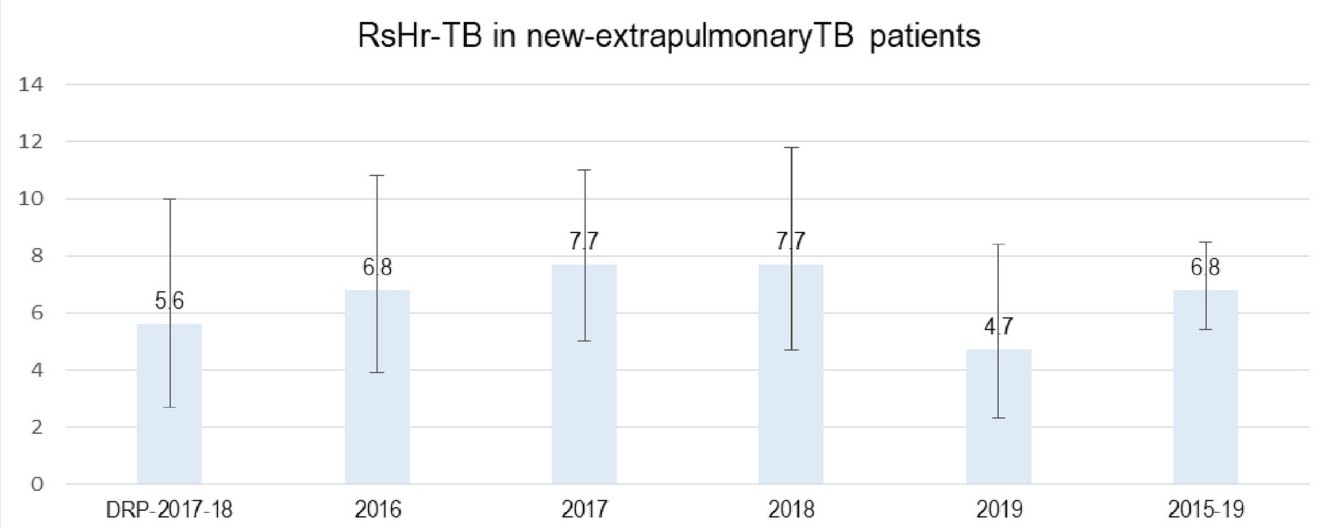

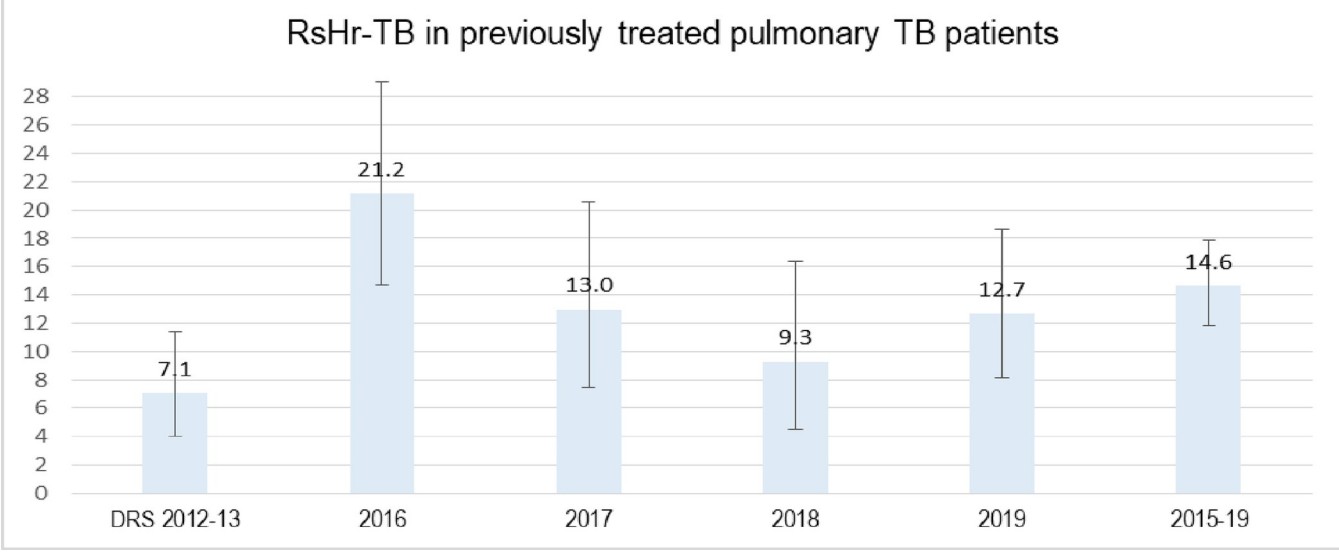

**Fig 2. Trend of isoniazid resistance in rifampicin sensitive new pulmonary TB, new extrapulmonary TB and previously treated pulmonary TB patients, National TB reference laboratory, Pakistan, 2015–19.** RsHr; Rifampicin sensitive isoniazid resistant.

**Table 4. Molecular characterization of isoniazid resistance in rifampicin resistant and sensitive Mtb isolates from pulmonary and extrapulmonary TB patients, National TB reference laboratory Pakistan 2015–19.**

| | All-TB | | RrHr-TB | | RsHr-TB | | RrHr-TB | | | | RsHr-TB | | | |
| --- | --- | --- | --- | --- | --- | --- | --- | --- | --- | --- | --- | --- | --- | --- |
| | | | | | | | PTB | | EPTB | | PTB | | EPTB | |
| INH resistant isolates | N = 4542 | | N = 4078 | | N = 464 | | N = 3971 | | N = 107 | | N = 389 | | N = 75 | |
| gWTpNWT | 660 | 14.5% | 528 | 12.9% | 132 | 28.4% | 519 | 13.1% | 9 | 8.4% | 115 | 29.6% | 17 | 22.7% |
| Any Mutation | 3882 | 85.5% | 3550 | 87.1% | 332 | 71.6% | 3452 | 86.9% | 98 | 91.6% | 274 | 70.4% | 58 | 77.3% |
| katG_S315T | 3213 | 70.7% | 3028 | 74.3% | 185 | 39.9% | 2943 | 74.1% | 85 | 79.4% | 153 | 39.3% | 32 | 42.7% |
| katG_315 | 90 | 2.0% | 84 | 2.1% | 6 | 1.3% | 84 | 2.1% | | 0.0% | 5 | 1.3% | 1 | 1.3% |
| inhA_c-15t | 411 | 9.0% | 287 | 7.0% | 124 | 26.7% | 277 | 7.0% | 10 | 9.3% | 101 | 26.0% | 23 | 30.7% |
| inhA_-15 | 17 | 0.4% | 8 | 0.2% | 9 | 1.9% | 8 | 0.2% | | 0.0% | 9 | 2.3% | | 0.0% |
| inhA_t-8c | 10 | 0.2% | 8 | 0.2% | 2 | 0.4% | 8 | 0.2% | | 0.0% | 1 | 0.3% | 1 | 1.3% |
| inhA_-8 | 5 | 0.1% | 3 | 0.1% | 2 | 0.4% | 2 | 0.1% | 1 | 0.9% | 2 | 0.5% | | 0.0% |
| inhA_t-8a | 6 | 0.1% | 3 | 0.1% | 3 | 0.6% | 3 | 0.1% | | 0.0% | 2 | 0.5% | 1 | 1.3% |
| inhA_-15,-8 | 1 | 0.0% | 1 | 0.0% | 0 | 0.0% | 1 | 0.0% | | 0.0% | | 0.0% | | 0.0% |
| inhA_c-15t & t-8c | 1 | 0.0% | 1 | 0.0% | 0 | 0.0% | 1 | 0.0% | | 0.0% | | 0.0% | | 0.0% |
| katG_S315T inhA_c-15t | 84 | 1.8% | 84 | 2.1% | 0 | 0.0% | 84 | 2.1% | | 0.0% | | 0.0% | | 0.0% |
| katG_S315T inhA_t-8c | 27 | 0.6% | 27 | 0.7% | 0 | 0.0% | 25 | 0.6% | 2 | 1.9% | | 0.0% | | 0.0% |
| katG_S315T inhA_t-8a | 4 | 0.1% | 4 | 0.1% | 0 | 0.0% | 4 | 0.1% | | 0.0% | | 0.0% | | 0.0% |
| katG_315 inhA_c-15t | 6 | 0.1% | 5 | 0.1% | 1 | 0.2% | 5 | 0.1% | | 0.0% | 1 | 0.3% | | 0.0% |
| katG_S315T inhA_-15 | 3 | 0.1% | 3 | 0.1% | 0 | 0.0% | 3 | 0.1% | | 0.0% | | 0.0% | | 0.0% |
| katG_S315T inhA_-8 | 3 | 0.1% | 3 | 0.1% | 0 | 0.0% | 3 | 0.1% | | 0.0% | | 0.0% | | 0.0% |
| katG_S315T inhA_a-16g | 1 | 0.0% | 1 | 0.0% | 0 | 0.0% | 1 | 0.0% | | 0.0% | | 0.0% | | 0.0% |
| **Summary** | | | | | | | | | | | | | | |
| gWTpNWT–n % | 660 | 14.5% | 528 | 12.9% | 132 | 28.4% | 519 | 13.1% | 9 | 8.4% | 115 | 29.6% | 17 | 22.7% |
| (95%CI) | (13.5–15.6) | | (11.9–14.0) | | (24.4–32.8) | | (12.0–14.2) | | ( 3.9–15.4) | | ( 25.1–34.4) | | (13.8–33.8) | |
| p-value | | | <0.01 | | | | 0.153 | | | | 0.225 | | | |
| katG mutation-n % | 3303 | 72.7% | 3112 | 76.3% | 191 | 41.2% | 3027 | 76.2% | 85 | 79.4% | 158 | 40.6% | 33 | 44.0% |
| (95%CI) | (71.4–74.0) | | (75.0–77.6) | | (36.6–45.8) | | (74.9–77.5) | | (70.5–86.6) | | (35.7–45.7) | | (35.5–55.9) | |
| p-value | | | <0.01 | | | | 0.443 | | | | 0.584 | | | |
| inhA mutation- n % | 451 | 9.9% | 311 | 7.6% | 140 | 30.2% | 300 | 7.6% | 11 | 10.3% | 115 | 29.6% | 25 | 33.3% |
| 95%CI | (9.1–10.8) | | (6.8–8.5) | | (26.0–34.4) | | (6.8–8.4) | | (5.2–17.7) | | (25.1–34.3) | | (22.9–45.2) | |
| p-Value | | | <0.01 | | | | 0.300 | | | | 0.523 | | | |
| Double Mutation- n % | 128 | 2.8% | 127 | 3.1% | 1 | 0.2% | 125 | 3.1% | 2 | 1.9% | 1 | 0.3% | 0 | 0.0% |
| 95%CI | (2.4–3.3) | | (2.6–3.7) | | (0.0–1.2) | | (2.6–3.7) | | (0.2–6.6) | | ( 0.007–1.4) | | (0.00–46.9) | |
| p-value | | | 0.187 | | | | 0.478 | | | | 0.129 | | | |

RrHr-Rifampicin resistant Isoniazid resistant, RsHr-Rifampicin sensitive Isoniazid resistant

new and previously treated (S5 Table) and annual trends. A more stable annual trend was seen for *katG* compared to *inhA* mutations. (Fig 3; S6 Table).

## Associated levofloxacin and pyrazinamide resistance

Among RMP resistant PTB isolates, resistance to LFX and PZA was reported respectively in 47.7% (95%CI: 46.2–49.2) and 44.8% (95%CI: 43.2–46.4) compared to 14.4% (13.3–15.7) and 3.7%, (95%CI: 3.0–4.5) in RMP sensitive. Similar pattern was seen in EPTB (Table 5; S7 Table).

**Rifampicin resistant TB.** Among RrHr-PTB isolates(n = 3969), resistance to LFX, PZA and combined LFX and PZA was 49%, 47% and 27.9%, which was higher compared to

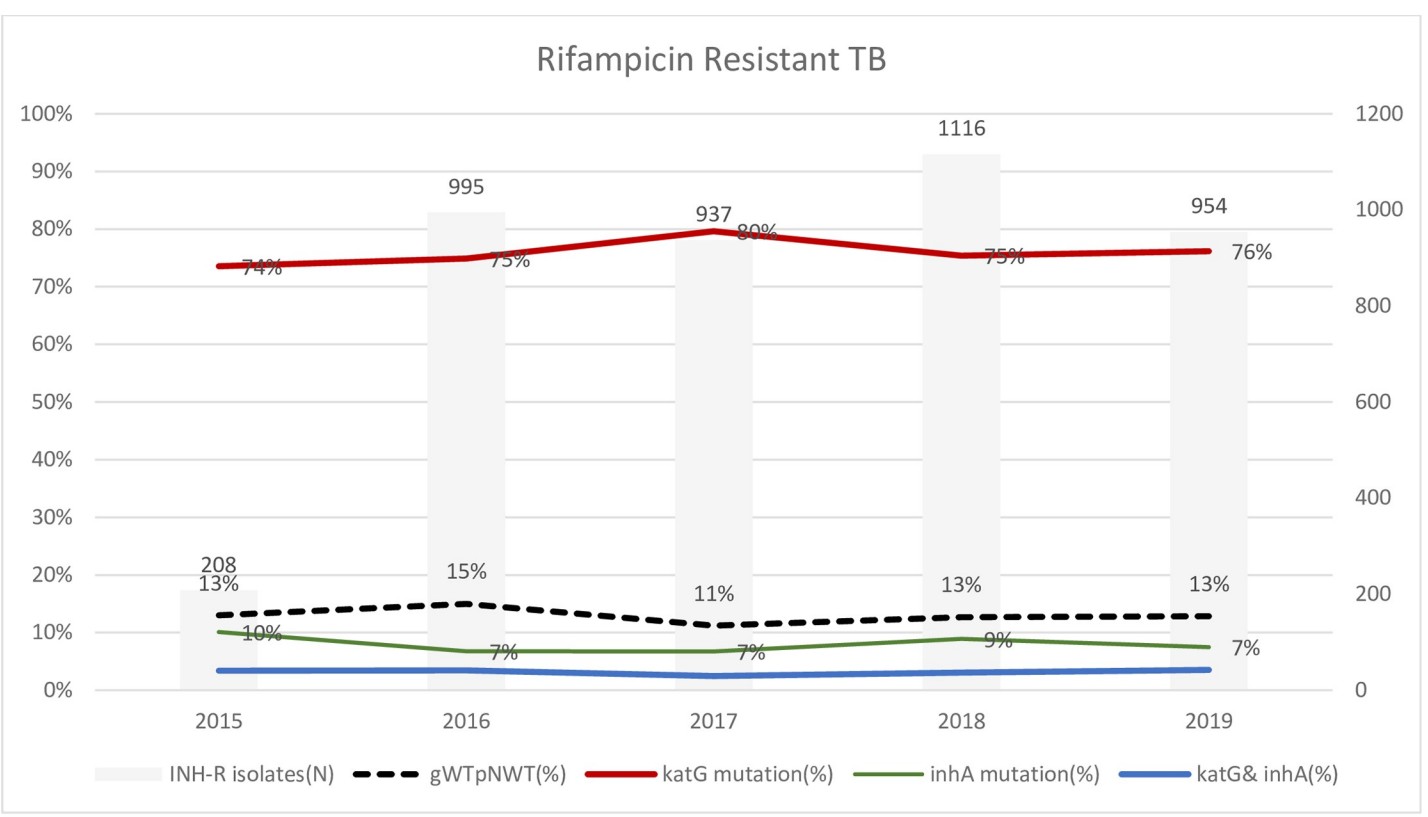

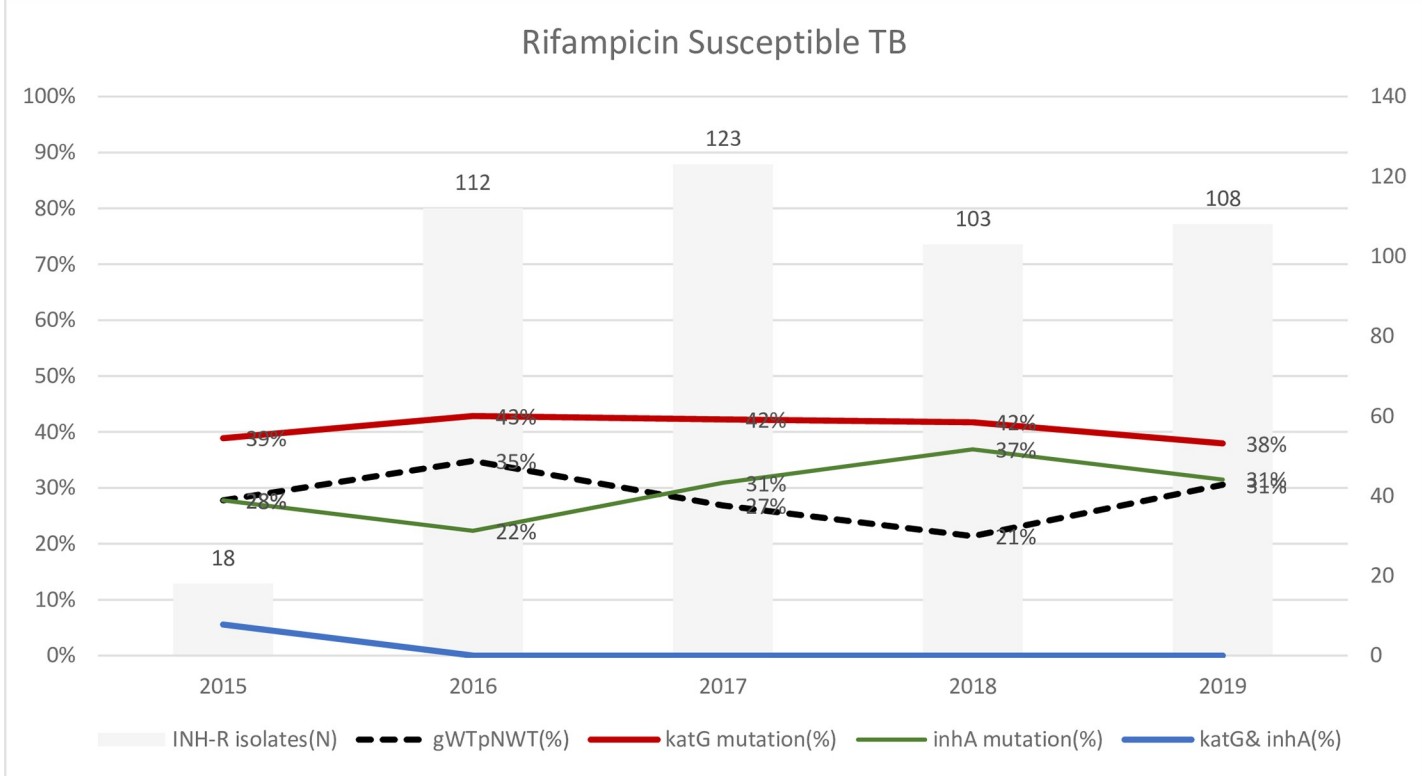

**Fig 3. Trend of isoniazid resistance profiles associated with rifampicin resistant and rifampicin sensitive isolates, National TB reference laboratory, Pakistan 2015–19.**

**Table 5. Levofloxacin and pyrazinamide resistance associated with isoniazid resistance in rifampicin resistant and sensitive *Mycobacterium Tuberculosis* isolates from pulmonary and extrapulmonary Tuberculosis patients, National TB reference laboratory 2015–2019.**

| Isoniazid resistance profile | Rifampicin Resistant TB | | | | | Rifampicin Sensitive TB | | | | |
|---|---|---|---|---|---|---|---|---|---|---|
| | Pulmonary TB | | Extrapulmonary TB | | p-value | Pulmonary TB | | Extrapulmonary TB | | p-value |
| | n/N | %(95%CI) | n/N | %(95% CI) | | n/N | %(95% CI) | n/N | %(95% CI) | |
| **Levofloxacin Resistance** | | | | | | | | | | |
| All Isolates | 1997/ 4183 | 47.7 (46.2– 49.2) | 53/ 118 | 44.9 (35.7– 54.3) | 0.548 | 486/ 3365 | 14.4 (13.3– 15.7) | 80/ 1117 | 7.2 (5.7– 8.8) | **<0.001** |
| H sensitive (gWTpWT) | 51/214 | 23.8 (18.3– 30.1) | 1/11 | 9.1(0.2– 41.3) | 0.259 | 388/ 2976 | 13.0 (11.8– 14.3) | 75/ 1042 | 7.2 (5.7– 8.9) | **<0.001** |
| H Resistant-All | 1946/ 3969 | 49.0 (47.5– 50.6) | 52/ 107 | 48.6 (38.8– 58.5) | 0.935 | 98/ 389 | 25.2 (21.0– 29.8) | 5/75 | 6.7(2.3– 14.9) | **<0.001** |
| ■ gWTpNWT | 213/ 518 | 41.1 (36.8– 45.4) | 5/9 | 55.6 (21.2– 86.3) | 0.381 | 38/ 115 | 33.0 (24.6– 42.4) | 1/17 | 5.9(0.01– 28.7) | 0.002 |
| ■ *inhA* mutation | 138/ 300 | 46.0 (40.3– 51.8) | 2/11 | 18.2(2.3– 51.8) | 0.069 | 24/ 115 | 20.9 (13.9– 29.4) | 1/25 | 4.0(0.1– 20.4) | 0.046 |
| ■ *katG* mutation | 1504/ 3026 | 49.7 (47.9– 51.5) | 44/ 85 | 51.8 (40.7– 62.7) | 0.703 | 36/ 158 | 22.8 (16.5– 30.1) | 3/33 | 9.1(1.9– 24.3) | 0.076 |
| ■ **Double Mutation** | **91/125** | **72.8 (64.1– 80.4)** | **1/2** | **50.0 (1.2– 98.7)** | 0.474 | 0/1 | 0.0(0.0– 97.5) | 0/0 | | NA |
| **Pyrazinamide Resistance** | | | | | | | | | | |
| All Isolates | 1655/ 3696 | 44.8 (43.2– 46.4) | 58/ 104 | 55.8 (45.7– 65.5) | 0.026 | 114/ 3048 | 3.7(3.0– 4.5) | 29/ 1019 | 2.8 (1.9– 4.1) | 0.174 |
| H sensitive (gWTpWT) | 13/202 | 6.4(3.5– 10.8) | 0/10 | 0.0 (0.0– 30.8) | 0.406 | 72/ 2704 | 2.7 (2.1– 3.3) | 23/ 951 | 2.4 (1.5– 3.6) | 0.618 |
| H Resistant- All | 1642/ 3494 | 47.0 (45.3– 48.7) | 58/ 94 | 61.7 (51.1– 71.5) | 0.005 | 42/ 344 | 12.2(8.9– 16.1) | 6/68 | 8.8 (3.3– 18.2) | 0.424 |
| ■ gWTpNWT | 133/ 445 | 29.9 (25.7– 34.4) | 2/6 | 33.3(4.3– 77.7) | 0.857 | 16/99 | 16.2(9.5– 24.9) | 1/15 | 6.7 (0.2– 31.9) | 0.336 |
| ■ *inhA* mutation | 84/266 | 31.6 (26.0– 37.5) | 4/10 | 40.0 (12.2– 73.8) | 0.576 | 5/105 | 4.8(1.6– 10.8) | 5/24 | 20.8(7.1– 42.2) | 0.008 |
| ■ *katG* mutation | **1354/ 2676** | **50.6 (48.7– 52.5)** | 51/ 76 | 67.1 (55.4– 77.5) | 0.005 | 21/ 139 | 15.1(9.6– 22.1) | 0/29 | 0.0 (0.0) | 0.025 |
| ■ **Double Mutation** | **71/107** | **66.4 (56.6– 75.2)** | 1/2 | 50.0(1.3– 98.7) | 0.672 | 0/1 | 0.0(0.00– 97.5) | 0/0 | | NA |
| **Combined Levofloxacin and Pyrazinamide Resistance** | | | | | | | | | | |
| All Isolates | 981/ 3696 | 26.5 (25.1– 28.0) | 31/ 104 | 29.8 (21.2– 39.6) | 0.453 | 31/ 3047 | 1.0(0.6– 1.4) | 2/ 1018 | 0.2(0.03– 0.80) | 0.013 |
| H sensitive (gWTpWT) | 6/202 | 3.0(0.01– 6.4) | 0/10 | 0.0(0.0– 30.8) | 0.578 | 9/ 2703 | 0.3(0.2– 0.6) | 2/950 | 0.2(0.02– 0.8) | **<0.001** |

(*Continued*)

**Table 5.** (Continued)

| Isoniazid resistance profile | Rifampicin Resistant TB | | | | | | Rifampicin Sensitive TB | | | | | |
| --- | --- | --- | --- | --- | --- | --- | --- | --- | --- | --- | --- | --- |
| | Pulmonary TB | | Extrapulmonary TB | | p-value | | Pulmonary TB | | Extrapulmonary TB | | p-value | |
| | n/N | %(95%CI) | n/N | %(95% CI) | | | n/N | %(95% CI) | n/N | %(95% CI) | | |
| H Resistant- All | 975/ 3494 | 27.9 (26.4–29.4) | 31/ 94 | 33.0 (23.6–43.4) | 0.277 | | 22/ 344 | 6.4(4.1–1.0) | 0/68 | 0.0(0.0) | 0.032 | |
| ■ gWTpNWT | 56/445 | 12.6(9.6–16.3) | 1/6 | 16.7 (0.4–64.1) | 0.764 | | 10/99 | 10.1(5.0–17.8) | 0/15 | 0.0 (0.0) | 0.198 | |
| ■ *inhA* mutation | 55/266 | 20.7 (16.0–26.0) | 0/10 | 0.0(0.0–30.8) | 0.108 | | 1/105 | 1.0(0.02–5.1) | 0/24 | 0.0(0.0) | 0.623 | |
| ■ *katG* mutation | **805/ 2676** | **30.1 (28.3–31.9)** | 29/ 76 | 38.2 (27.2–50.0) | 0.130 | | 11/ 139 | 7.9(4.3–13.7) | 0/29 | 0.0 (0.0) | 0.117 | |
| ■ **Double Mutation** | **59/107** | **55.1 (45.2–64.8)** | 1/2 | 50.0 (12.6–98.7) | 0.886 | | 0/1 | 0.0(0.0–97.5) | 0/0 | | NA | |

n-number of clinical isolates resistant to Levofloxacin and/or Pyrazinamide, N-Number of Isolates tested, WT-wild type, g-genotypic, p-phenotypic

reported resistance in RrHs-TB (n = 214) of 23.8%, 6.4% and 3.0%. Similarly among RrHr-EPTB (n = 107) reported resistance of 48.6%, 61.7% and 33.0% was higher compared to 9.1%, 0% and 0% in RrHs-TB (n = 11) (Table 5).Within RrHr-TB, significant difference in resistance was not seen between PTB and EPTB (*P* = .08) (Table 5), new and previously treated TB (*P* = .956) (S7 Table) and annual trends (S8 Table).

**Rifampicin sensitive TB.** Among RsHr-PTB(n = 389), resistance to LFX, PZA and combined LFX and PZA, isolates was reported in 25.2%, 12.2% and 6.4% respectively compared to 13.0%, 3.7% and 1.0 in RsHs-PTB(n = 2976). Among RsHr-EPTB (N = 75)isolates, resistant was reported respectively in 6.7%, 8.8% and 0% compared to 7.2%, 2.8% and 0.2% in RsHs-EPTB (N = 1042).No significant changes were see in the annual trend of resistance to LFX and PZA. (S8 Table).

**Isoniazid conferring mutations and Levofloxacin and pyrazinamide resistance.**
Among RrHr-PTB isolates, a relative higher resistance to LFX was seen in isolates with double mutations and relative higher resistance to PZA and combined LFX and PZA resistance was seen associated with *katG* and double mutations. (Table 5; S7 Table; S1 Fig).

## Discussion

We analyzed a large DST data set from 11045 patients registered during 2015–2019 in NTRL, Pakistan. During the study period, pDST was performed on 87%, gDST on 92% and 79.5% of isolates were tested by both methods. We analyzed results of 8787 Mtb isolates including 1236 from EPTB patients, tested by both phenotypic and genotypic methods. Key findings of our analysis include significant difference in resistance detected by WHO recommended pDST (MGIT 960) and gDST (LPA) methods for both RMP and INH, a significantly lower sensitivity of gDST to detect INH resistance in RMP sensitive TB, difference in genetic profiles of INH resistance associated with RMP resistant and sensitive TB, a higher level of LFX and PZA resistance associated with INH resistance in RMP resistant and sensitive population. Additionally

we noted a stable annual trend of INH resistance in RMP sensitive new TB patients, INH resistance profiles, associated FQ and PZA resistance and INH resistance missed by gDST method.

We reported, an INH resistance of 9.8% (95%CI; 8.7–11.1) in new and 14.6% (95%CI: 11.8–17.9) in previously treated RMP sensitive PTB patients. Among new PTB, a higher point estimates compared to the estimates of DRS (8.3%, 95%CI: 7.0–10.7) conducted in 2013 [14] can be explained based on the possibility of a selected higher risk individuals among new RMP sensitive patients referred for testing in routine settings. However a stable RsHr resistance trend was seen during the study period with an insignificant increase only in 2019. Among previously treated, compared to DRS, the proportion of RsHr-TB was significantly higher but fluctuations were seen in annual trends. Fluctuation in resistance was most likely due to variation in proportion of relapse vs failures of previous treatment among patient tested. However, a higher RsHr reported among previously treated was consistent with findings of other similar studies from Pakistan [20–22]. Among children with PTB (n = 302), estimated RsHr-TB of 11.9%(8.5–16.1), was higher but not statistically different from adults(p = .189) and was consistent with the global estimates of 12.1% (95%CI:9.8% to 14.8%) among all childhood TB cases [23]. However prevalence estimates in children can also be argued as not being a true representative of childhood TB population as most likely specimens from sicker children having access to specialized TB care and diagnostic facilities were tested. Among RMP sensitive new EPTB patient (n = 1058), INH resistance of 6.8% (95%CI; 5.4–8.5) was significantly lower compared to PTB ($P < .001$) in same study population but was consistent with primary drug resistance (5.6%, 95CI; 2.7–10.0) reported in EPTB [19] and PTB in Pakistan [14, 20, 24]. Most of the EPTB patients in contrast to PTB, were referred for diagnosis of TB and only a few had history of previous TB treatment (<10%), with possibility of these estimates being a true reflection of RsHr in EPTB at population level.

Among Mtb isolates tested by both methods (n = 8787), discordance in RMP results was reported in 7.7% of all and 15.8% (678/4303) of RMP resistant isolates. Discordance in RMP results was also reported previously in random population-based DRS (2013) in which all samples were tested in parallel using Xpert MTB/Rif and Lowenstein Jensen (LJ) media and sequencing was performed to confirm and resolve discordance in RMP results [14]. However in DRS, discordance of 11.7% was reported among all RMP resistant isolates, including 4% (4/104) missed on pDST and 7.7% (8/104) by gDST. Contrary to DRS, in this study 13.2% (n = 570) were missed by MGIT (pDST) and 2.5% (n = 108) by LPA (gDST). The most plausible explanation for a higher number of observed missed RMP resistance by pDST (13.2% vs 4%) in study sample was due to use of automated liquid DST (MGIT), which is known to miss higher proportion of RMP resistant cases compared to LJ media [25, 26]. On the other hand the lower proportion of RMP resistant cases missed by gDST (7.7% vs 2.5%) was most likely an effect of the current diagnostic algorithm followed in programme settings in which patients reported RMP sensitive are initiated on standard first line treatment and are not investigated further unless they fail to respond or fail treatment.

Recently conducted systematic review and meta-analysis, reported significant higher failure rate among INH resistant compared to susceptible TB patients when treated with standard first-line drugs regimens [13]. Limitations of currently available pDST and gDST in detecting RMP resistance are well recognized [25–27]. Selection of either one of the two DST method for routine practice is likely to result in important diagnostic and clinical implications. In our study population, without genotypic DST, 10.1% (411/4078) of the MDR would have been reported as RsHr-TB and treatment in these instances with standard first line treatment regimen would likely have resulted in suboptimal treatment outcomes. In a recently published study, a non-negligible extent of misclassifying MDR-TB as INH-resistant TB is demonstrated and impact of treating patients with missed RMP resistance for RsHr-TB with WHO

recommended FQ containing regimen is strongly argued [27]. In another study from South Africa 15% of INH resistant isolates initially tested negative for RMP resistance by all three WHO-endorsed commercial tests were reclassified as MDR on identification of resistance conferring mutation (*rpoB* Ile491phe) using deep sequencing [28].

In our data set, contrary to RMP, a higher proportion of INH resistance (14.5%; n = 660) was missed by gDST. Our finding are consistent with the results of a recent study from eastern DRC, in which INH resistance was reported in only 55% of the RMP resistant cases detected by Xpert MTB/RIF on subsequent testing by LPA, raising an argument on use of RMP resistance as surrogate marker for MDR [29].

We studied 4542 INH resistant isolates for molecular markers and mutations causing INH resistance were identified in 85.5% with frequency of *katG*, *inhA* and combined *katG and inhA* mutations in 72.7%, 9.9% and 2.8% respectively. Our findings are consistent with the published data [11, 12, 30] but with a lower proportion of combined *katG* and *inhA* mutations in our population [12, 31]. INH resistance profiles when studied, stratified by RMP results, significant differences were reported between RrHr-TB (n = 4078) and RsHr-TB(n = 464) with regard to proportion of INH conferring mutation detected (87.1% vs 71.6%) and frequency of the mutations in *inhA* (7.6 vs 30.2%), *katG* (76.3 vs 41.2%) and combined *inhA* and *katG* (3.1% vs 0.2%). The distribution of INH resistance mutations among RsHr-TB is less well mapped globally, however our findings are consistent with the estimates from an international collection of over 5,000 strains with reported frequency of S315T *kat*G in 88.7% of MDR and 61.3% of RsHr-TB showing substantially higher representation of *katG* among MDR strains [32].

In our study population of RsHr-TB, 41% had mutation in *katG*, 30% in *inhA* and only one isolates had combined mutation. Published data on the influence of genotype on treatment outcomes of RsHr-TB are conflicting, study from Vietnam, suggest that *katG* mutations and not *inhA* are associated with unfavorable treatment outcomes [33]. Whereas in a study from South Africa, no evidence was found suggesting that specific isoniazid resistance conferring mutations are associated with poor treatment outcomes, and results showed that patients with *katG* mutations had greater odds of successful outcome when treated with high-dose isoniazid compared to those who received standard dose [34].Few studies have also evaluated the effectiveness of high-dose isoniazid in patients with DRTB and limited data available suggest clinical benefit without a higher toxicity [35, 36]. A recent study has demonstrated that high dose INH in MDR patients with *inhA* mutations has similar magnitude of bactericidal activity as with standard doses in drug-susceptible TB patients [37].

In Pakistan, one of the key challenge for treating MDR-TB and RsHr-TB is high FQ resistance. We reported a higher FQ resistance in RMP resistant and sensitive isolates compared to population level resistance [12, 15] but was consistent with previous laboratory based study from Pakistan [38]. In RMP resistant and sensitive population, significantly higher resistance to LFX and PZA was noted associated with INH resistance and was statistically higher for all INH resistance profiles. However a relative higher resistance was seen associated with combined *katG* and *inhA* mutations, consistent with findings of a recent study [31]. A high LFX resistance and lower frequency of *katG* mutations in RsHr-TB in our study population imply consideration for high dose INH rather than LFX as a treatment option.

The End TB Strategy released in 2015 calls for the early diagnosis of TB including universal DST [39]. Xpert MTB/RIF assay was endorsed by WHO in 2010, however even in 2018, among bacteriologically confirmed TB patients, only 51% globally and 45% in Pakistan were tested for RMP resistance [7]. Diagnostic and operational challenges to implement universal INH testing and treatment for INH resistant TB, are more complex, because of the larger population of RMP sensitive patients and complex laboratory capacity required for testing in the

absence of a rapid and convenient diagnostic platform for detection of INH resistance and challenges faced in decentralization of LPA in most of HBC settings. Additionally in Pakistan country context, even with systematic testing, 30 percent of the INH resistance are likely to be missed if tested by LPA only, second, there is lack of laboratory capacity to systematically exclude RMP resistance which are missed by Xpert MTB/Rif, third, high LFX resistance makes FQ testing mandatory for all and lastly, FQ resistance in a substantial number of RsHr-TB patients will render them ineligible for recommended treatment.

Currently HBC are struggling to develop capacity for comprehensive second line DST including new and repurpose drugs [5] and to establish sequencing capacity to diagnose RMP and other drug resistance not detected by routine tests. Plans to implement universal testing and treatment for INH resistant TB with existing capacity are likely to overwhelm laboratory systems at the cost of compromising services for RMP resistant TB patients. A new GeneXpert cartridge for testing INH and FQ resistance is expected to be available in near future [40, 41], however countries will need to invest in procurement of next generation modules and to make it available at same level of health care as RMP testing for detection of INH and FQ resistance in RMP sensitive TB patients in parallel with plans to implement treatment for INH resistance. In addition performance evaluation of new diagnostic tests in both RMP resistant and sensitive population in different geographical settings also needs consideration.

The value of any such laboratory study would be greatly increased if treatment outcomes could be linked to the resistance profile. Studies are also needed to fill knowledge gap in isoniazid acetylator status of the population in HBC and to evaluate effectiveness, optimal dosing and potential toxicity of high-dose isoniazid to offer simple treatment options applicable in program settings.

We studied retrospective data and possibility of errors in routinely collected patient information cannot be excluded. Furthermore findings of this study cannot be generalized to the population level. However laboratory based surveillance can complement random survey as analysis provides information on drug resistance pattern in a large dataset of DRTB patients at time of treatment initiation and can guide in country specific planning for TB diagnostics, diagnostic algorithm and appropriate treatment regimens for drug resistance in TB patients.

## Supporting information

**S1 Table. Demographic and clinical characteristics of pulmonary and extrapulmonary tuberculosis patients tested for drug susceptibility at national TB reference laboratory Pakistan, 2015–2019.**
(PDF)

**S2 Table. Specimen types from pulmonary and extrapulmonary tuberculosis patients processed for drug susceptibility testing, national TB reference laboratory Pakistan, 2015–19.**
(PDF)

**S3 Table. Annual trend and difference in isoniazid and rifampicin resistance detected by phenotypic and genotypic DST methods, in pulmonary and extrapulmonary tuberculosis patients, national TB reference laboratory Pakistan 2015–19.** R-rifampicin, H-isoniazid, r-resistant, s-susceptible, DST-drug susceptibility testing, p-phenotypic; g-genotypic.
(PDF)

**S4 Table. Rifampicin and isoniazid resistance trend and correlation between phenotypic and genotypic drug susceptibility testing, national TB reference Laboratory, Pakistan 2015–2019.** RMP-rifampicin, INH-isoniazid, R-resistant, S-sensitive, pDST-phenotypic, drug

susceptibility testing; gDST-Genotypic drug susceptibility testing.
(PDF)

**S5 Table. Molecular characterization of isoniazid resistance in Mtb isolates from pulmonary and extrapulmonary TB patients, stratified by rifampicin resistance and history of TB treatment, national TB reference laboratory, Pakistan 2015–2019.** INH-isoniazid.
(PDF)

**S6 Table. Annual trend of genotypic profiles of isoniazid resistance associated with rifampicin resistant and sensitive Mtb isolates from new and previously treated TB patients, national TB reference laboratory, Pakistan, 2015–19.** RMP-rifampicin, INH-isoniazid, PT-Previously treated.
(PDF)

**S7 Table. Levofloxacin and pyrazinamide resistance associated with isoniazid resistance in rifampicin resistant and sensitive Mtb isolates from pulmonary and extra pulmonary TB patient, stratified by history of TB treatment, national TB reference laboratory Pakistan, 2015–19.** n-Number of isolates resistant to Levofloxacin and/or Pyrazinamide, N- Number of isolates tested INH-isoniazid.
(PDF)

**S8 Table. Annual trend of levofloxacin and pyrazinamide resistance associated with isoniazid resistance in rifampicin resistant and sensitive Mtb isolates from pulmonary and extrapulmonary TB patients, national TB reference laboratory Pakistan, 2015–19.**
LFX-Levofloxacin, PZA-Pyrazinamide, n = Number of isolates resistant, N = Number of isolates tested.
(PDF)

**S1 Fig. Levofloxacin, pyrazinamide and combined levofloxacin and pyrazinamide resistance associated with isoniazid sensitive, isoniazid resistant and specific isoniazid conferring mutations in rifampicin resistant and sensitive isolates from pulmonary and extrapulmonary TB patients.** National TB reference Laboratory, Pakistan 2015–2019. Rr-Rifampicin resistant, Rs-Rifampicin sensitive, Hr-Isoniazid resistance, Hs-Isoniazid sensitive, isoniazid sensitive.
(TIF)

## Acknowledgments

We would like to greatly acknowledge contribution of The global fund to establish and expand the National laboratory network capacity for bacteriology, drug susceptibility testing and surveillance of antituberculosis drug resistance in Pakistan.

## Author Contributions

**Conceptualization:** Sabira Tahseen, Faisal Masood Khanzada, Aurangzaib Quadir Baloch, Tehmina Mustafa.

**Data curation:** Sabira Tahseen, Faisal Masood Khanzada.

**Formal analysis:** Sabira Tahseen, Faisal Masood Khanzada, Tehmina Mustafa.

**Investigation:** Faisal Masood Khanzada, Alamdar Hussain Rizvi, Mahmood Qadir, Aisha Ghazal.

**Methodology:** Sabira Tahseen, Faisal Masood Khanzada, Tehmina Mustafa.

**Project administration:** Sabira Tahseen.

**Resources:** Aurangzaib Quadir Baloch.

**Supervision:** Sabira Tahseen, Faisal Masood Khanzada, Alamdar Hussain Rizvi, Tehmina Mustafa.

**Validation:** Sabira Tahseen, Faisal Masood Khanzada.

**Visualization:** Sabira Tahseen.

**Writing – original draft:** Sabira Tahseen.

**Writing – review & editing:** Sabira Tahseen, Faisal Masood Khanzada, Alamdar Hussain Rizvi, Mahmood Qadir, Aisha Ghazal, Aurangzaib Quadir Baloch, Tehmina Mustafa.

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
