## [Decision Letter · Decision Letter 0]

20 Aug 2020

PONE-D-20-21797

Phenotypic and genotypic profile of isoniazid resistance in pulmonary and extrapulmonary tuberculosis in Pakistan; A retrospective study of laboratory based surveillance 2015-19

PLOS ONE

Dear Dr. Tahseen,

Thank you for submitting your manuscript to PLOS ONE. After careful consideration, we feel that it has merit but does not fully meet PLOS ONE’s publication criteria as it currently stands. Therefore, we invite you to submit a revised version of the manuscript that addresses the points raised during the review process.

We look forward to receiving your revised manuscript.

Kind regards,

Hasnain Seyed Ehtesham

Academic Editor

PLOS ONE

Journal Requirements:

2. The study design section in your manuscript states that “This is a retrospective five-year (January 2015-December 2015) laboratory based surveillance study”. Please check that these dates are correct.

Additional Editor Comments (if provided):

Minor Revision

Reviewers' comments:

Reviewer's Responses to Questions

**Comments to the Author**

1. Is the manuscript technically sound, and do the data support the conclusions?

Reviewer #1: Yes

Reviewer #2: Yes

2. Has the statistical analysis been performed appropriately and rigorously? 

Reviewer #1: Yes

Reviewer #2: Yes

3. Have the authors made all data underlying the findings in their manuscript fully available?

Reviewer #1: No

Reviewer #2: Yes

4. Is the manuscript presented in an intelligible fashion and written in standard English?

Reviewer #1: Yes

Reviewer #2: Yes

5. Review Comments to the Author

Reviewer #1: The manuscript reflects a large retrospective 5 year study of INH resistance in defined provinces of Pakistan and thus addresses gaps in understanding the factors that identify INHresistance, the effect on mising out INHand RFP resistance and the connection of INHand RFP resistance patterns to fluoroquinolones and PZA. In my opinion, the manuscript provides valuable comprehensive regional information.

The manuscript is well drafted and the discussion is well constructed. However there are multiple minor edits that are needed all through the manuscript. Attention is drawn to a typo error in line 124, wrt the 5 year duration. December 2015 should be replaced with December 2019 and to the subheading Discussion (not Discussions)

While undoubtedly the manuscript focusses on INH, there is a constant comparison of findings of RFP resistance patterns. Might it be warranted that RFP should also find a place in the title.

The laboratory methods need some more detailing in light of the discussion line 328 to 332 where mention is made of using Gene XpertMTB/Rif and WGS to confirm all RFP resistant cases. If this was undertaken with isolates from the present study, then this confirmation should be mentioned briefly in the Laaboratory Methods section citing the reference.

Sixty four percent of the patients tested here were retreateated patients. and 45% had previous treatment regimen for new patients. Therefore amplification of resistance is likely to have taken place. Would the authors like to speculate why INH resistance is sizeable also in new INHcases

In fig.2 RsHR TB in pulmonary TB patients, there is a significant drop between 2016 and 2017 aven 2018. Is there a reason for this trend.

It would be nice to know the annual variation if any in the proportion of inh and katG mutations in the different strata of patients. Is the proportion of katG mutations overall increasing temporally in the bacterial population?

The argument for programme adoption of genotypic technologies starting from Xpert like molecular testing platform for INH and extending to WGS and deep sequencing for confirming RFP and more so is an apt one and constitutes an important message of the manuscript. However the authors should attempt to shorten the lengthy discussion without omitting the essential points for enhancing readibility.

The value of such a laboratory study would be greatly increased if treatment outcomes could be linked to the genotypic and phenotypic DR findings. This could be a message for the design of future studies.

Reviewer #2: The retrospective study addresses an important aspect of TB control. However, the discussion needs to be more focussed, relevant to the results obtained. Statements like "It is critical to have a simple, accurate (>90% sensitivity) and affordable test to rapidly detect INH and FQ resistance" are sweeping statements, that state the obvious, and could be avoided.

6. PLOS authors have the option to publish the peer review history of their article (what does this mean?). If published, this will include your full peer review and any attached files.

Reviewer #1: No

Reviewer #2: No

---

## [Author Response · Author response to Decision Letter 0]

28 Aug 2020

Dear editor, 

We wish to thank you for thorough review and constructive comments on our manuscript, 

PONE-D-20-21797

Phenotypic and genotypic profile of isoniazid resistance in pulmonary and extrapulmonary tuberculosis in Pakistan; A retrospective study of laboratory based surveillance 2015-19

(Original Article)

We have revised the manuscript based on each point raised by the academic editor and reviewers Here we provide point-to-point replies. The changes in the paper are marked in red color in the resubmitted manuscript. 

Comments to the Author

Response: The manuscript is formatted according to style requirement of PLOS One 

2. The study design section in your manuscript states that “This is a retrospective five-year (January 2015-December 2015) laboratory based surveillance study”. Please check that these dates are correct.

Response: This is five year study (January 2015-December 2019), typing error is corrected in revised version

Review Comments to the Author

Reviewer #1: The manuscript reflects a large retrospective 5 year study of INH resistance in defined provinces of Pakistan and thus addresses gaps in understanding the factors that identify INH resistance, the effect on missing out INH and RFP resistance and the connection of INH and RFP resistance patterns to fluoroquinolones and PZA. In my opinion, the manuscript provides valuable comprehensive regional information.

Response: Thank you very much for your comments 

Comment: The manuscript is well drafted and the discussion is well constructed. However there are multiple minor edits that are needed all through the manuscript. Attention is drawn to a typo error in line 124, wrt the 5 year duration. December 2015 should be replaced with December 2019 and to the subheading Discussion (not Discussions)

Response: Thank you very much, we have made two correction advised along with other edits. 

Comment: While undoubtedly the manuscript focusses on INH, there is a constant comparison of findings of RFP resistance patterns. Might it be warranted that RFP should also find a place in the title

Response: Thank you for your advice, we totally agree and have changed the title to 

“Isoniazid resistance profile and associated levofloxacin and pyrazinamide resistance in rifampicin resistant and sensitive isolates from pulmonary and extrapulmonary tuberculosis patients in Pakistan: A laboratory based surveillance study 2015-19”

Comment: The laboratory methods need some more detailing in light of the discussion line 328 to 332 where mention is made of using GeneXpert MTB/Rif and WGS to confirm all RFP resistant cases. If this was undertaken with isolates from the present study, then this confirmation should be mentioned briefly in the Laboratory Methods section citing the reference.

Response: In drug resistance survey conducted in 2013(reference 14), sequencing was performed for all rifampicin isolates in SRL –Antwerp Belgium. However sequencing was not performed for discordance reported in routine settings as facilities were not established. 

In Pakistan GeneXpert services are decentralized and in most of the cases, samples from known rifampicin resistance cases are referred for comprehensive drug susceptibility testing. We have added a small description under study setting. (row 95-98 ) and laboratory methods(row 117-118) to clarify.

Comment: Sixty four percent of the patients tested here were retreateated patients and 45% had previous treatment regimen for new patients. Therefore amplification of resistance is likely to have taken place. Would the authors like to speculate why INH resistance is sizeable also in new INH cases.

Response: Unlike rifampicin resistant patients, only selected new rifampicin sensitive patient are referred for testing, we have elaborated this in discussion (row 290-292)

Comment: In fig.2 RsHR-TB in pulmonary TB patients, there is a significant drop between 2016 and 2017 even 2018. Is there a reason for this trend?

Response: Yes contrary to new TB patients, there is fluctuation in annual resistance trend among previously treated. Trend analysis for previously treated was done only for those with history of Cat-1with possibility of variation in proportion patient with different treatment outcomes of previous treatment eg relapse vs failures among patient included affecting the resistance pattern. As annual number of these patients is small, fluctuation in point estimates are seen with wide confidence intervals. We have added small description for possible reasons (Row 298-300) in discussion.

Comment: It would be nice to know the annual variation if any in the proportion of inhA and katG mutations in the different strata of patients. Is the proportion of katG mutations overall increasing temporally in the bacterial population?

Response: We have added more data in s2-Table to show annual trend in INH resistance and genetic profile for new and previously treated patients. However spike in katG in any of the strata is not noted. 

Comment: The argument for programme adoption of genotypic technologies starting from Xpert like molecular testing platform for INH and extending to WGS and deep sequencing for confirming RFP and more so is an apt one and constitutes an important message of the manuscript. However the authors should attempt to shorten the lengthy discussion without omitting the essential points for enhancing readability.

The value of such a laboratory study would be greatly increased if treatment outcomes could be linked to the genotypic and phenotypic DR findings. This could be a message for the design of future studies. 

Response: Thank you for your comments and advise; we have shorten the discussion, re-arranged text to align it with flow of the result section and made it more readable and added suggestion of linking resistance profile with treatment outcomes in paragraph on future studies.(Row 395-398)

Reviewer #2: 

Comment: The retrospective study addresses an important aspect of TB control. However, the discussion needs to be more focused, relevant to the results obtained. Statements like "It is critical to have a simple, accurate (>90% sensitivity) and affordable test to rapidly detect INH and FQ resistance" are sweeping statements, that state the obvious, and could be avoided.

Response: Thank you very much for your comment, we have now edited the discussion section, shorten it and aligned it with results section. We have added a new reference and press release (16 July ’20) on newly launched XDR cartridge (detection of fluoroquinolone, INH, SLI) in discussion while shortening viewpoints on risks and challenges for implementing INH testing and treatment. 

Summary of changes in manuscript; 

• All changes are marked in red color in the resubmitted manuscript. Changes marked in red also include text which has been rearranged without changing the content to improve readability. 

• An error was identified in Figure 2, that has been corrected and figure is replaced with new one.

• Two reference including a recent press release (16 July ’20) on newly launched XDR cartridge (detection of fluoroquinolone, INH, SLI) are added and viewpoints on new development added in the discussion.

• New data is added in S2 Table based on reviewer #1 advice

• Spelling mistake in one of the co-author’s name is corrected

---

## [Editor Report · Decision Letter 1]

4 Sep 2020

Isoniazid resistance profile and associated levofloxacin and pyrazinamide resistance in rifampicin resistant and sensitive isolates from pulmonary and extrapulmonary tuberculosis patients in Pakistan: A laboratory based surveillance study 2015-19

PONE-D-20-21797R1

Dear Dr. Tahseen,

We’re pleased to inform you that your manuscript has been judged scientifically suitable for publication and will be formally accepted for publication once it meets all outstanding technical requirements.

Kind regards,

Hasnain Seyed Ehtesham

Academic Editor

PLOS ONE

Additional Editor Comments (optional):

I have gone through this revised manuscript and also the Authors response to the comments of both the reviewers. This manuscript was sent for minor revision and Authors have made the necessary changes addressing the comments of both the reviewers. All issues have been addressed. Error in the fig 2 has been corrected and another figure is replaced with the new one. Authors have shortened the discussion part and re-arranged the text to provide more clarity to the readers.

I recommend this manuscript for publication.
---

## [Editor Report · Acceptance letter]

14 Sep 2020

PONE-D-20-21797R1 

Isoniazid resistance profile and associated levofloxacin and pyrazinamide resistance in rifampicin resistant and sensitive isolates from pulmonary and extrapulmonary tuberculosis patients in Pakistan: A laboratory based surveillance study 2015-19 

Dear Dr. Tahseen:

I'm pleased to inform you that your manuscript has been deemed suitable for publication in PLOS ONE. Congratulations! Your manuscript is now with our production department. 

Kind regards, 

on behalf of

Prof Hasnain Seyed Ehtesham 

Academic Editor

PLOS ONE